# Clindamycin Efficacy for *Cutibacterium acnes* Shoulder Device-Related Infections

**DOI:** 10.3390/antibiotics11050608

**Published:** 2022-04-30

**Authors:** Audrey Courdurié, Romain Lotte, Raymond Ruimy, Vincent Cauhape, Michel Carles, Marc-Olivier Gauci, Pascal Boileau, Johan Courjon

**Affiliations:** 1Infectious Diseases Department, Université Côte d’Azur, CHU Nice, 06202 Nice, France; courdurie.a@chu-nice.fr (A.C.); cauhape.v@chu-nice.fr (V.C.); carles.m@chu-nice.fr (M.C.); 2Department of Bacteriology, Université Côte d’Azur, CHU Nice, 06202 Nice, France; lotte.r@chu-nice.fr (R.L.); ruimy.r@chu-nice.fr (R.R.); 3INSERM, C3M, 06204 Nice, France; 4Institut Universitaire Locomoteur & Sport (IULS), Unité de Recherche Clinique Côte d’Azur (UR2CA), Hôpital Pasteur 2 CHU de Nice, Université Côte D’Azur, 06000 Nice, France; gauci.mo@chu-nice.fr; 5ICR-Institut de Chirurgie Réparatrice Locomoteur et Sports–Groupe KANTYS, 06000 Nice, France; boileau.pascal@wanadoo.fr

**Keywords:** *Cutibacterium acnes*, shoulder implant, clindamycin

## Abstract

Clindamycin is an antibiotic with high bioavailability and appropriate bone diffusion, often proposed as an alternative in guidelines for *C. acnes* prosthetic joint infections. We aimed to evaluate the efficacy of clindamycin in the treatment of *C. acnes* shoulder implant joint infections (SIJI). Methods: A retrospective analysis was conducted at the University Hospital of Nice (France) between 2010 and 2019. We included patients with one shoulder implant surgical procedure and at least one *C. acnes* positive sample. We selected the *C. acnes* SIJI according to French and international recommendations. The primary endpoint was favorable outcome of *C. acnes* SIJI treatment after at least 1-year follow-up in the clindamycin group compared to another therapeutic group. Results: Forty-eight SIJI were identified and 33 were treated with clindamycin, among which 25 were treated with monotherapy. The median duration of clindamycin antibiotherapy was 6 weeks. The average follow-up was 45 months; one patient was lost to follow-up. Twenty-seven patients out of 33 (82%) were cured with clindamycin, compared to 9/12 (75%) with other antibiotics. The rate of favorable outcomes increased to 27/31 (87%) with clindamycin and to 9/10 (90%) for other antibiotics when no septic revision strategies were excluded (*P* = 1.00). Conclusions: The therapeutic strategy based on one- or two-stage revision associated with 6 weeks of clindamycin seems to be effective.

## 1. Introduction

Due to the increase in the average age of the world population and the evolution of surgical techniques, the number of shoulder prosthesis (SP) procedures continues to increase. According to the French Health Authority, 12,500 SP procedures were performed in France in 2014 [1]. Prosthetic joint infections (PJI) occur in 2% of SP procedures [2]. Infection is considered to be the most serious complication that leads to functional impairment; moreover, it has a significant economic impact [3,4]. The bacteria that are the most often identified during shoulder prosthetic joint infections (SPJI) are commensal microorganisms of the skin such as coagulase-negative staphylococci (CoNS) and *Cutibacterium acnes,* the latter representing nearly 40% of SPJI [5,6]. *C. acnes* is a Gram-positive commensal bacillus found mostly on the skin and especially in pilo-sebaceous follicles of the upper body such as the shoulders [7]. The diagnosis of infections caused by low virulence organisms such as *C. acnes* often represents a diagnostic challenge for the surgeon, since the symptoms are often mild and the usual diagnostic tools have a low level of sensitivity due to a weak inflammatory response [8].

In order to treat *C. acnes* SPJI, several antibiotics are available. The Infectious Diseases Society of America recommends type A penicillin as a first choice [9], which is always active on *C. acnes*, but the saturable absorption rate of amoxicillin calls into question its use for oral antibiotic treatment of PJI [10]. The susceptibility profile of *C. acnes* in our clinic also allows the use of fluoroquinolones. This antibiotic family shows a good bioavailability together with an extensive bone diffusion, but significantly impairs the intestinal microbiota and leads to safety issues during PJI treatment [11,12,13]. Rifampicin is also known for its high bioavailability and antibiofilm activity, but needs to be used in combination; it also displays tolerance issues and several drug interactions [12,14]. The limitations of these different molecules led us to prioritize the use of clindamycin in our hospital.

Clindamycin is recommended as an alternative monotherapy choice by the 2018 Philadelphia International Consensus and the Infectious Diseases Society of America [9,15]. Clindamycin is an appropriate molecule for bone and joint infections (BJI) because of its significant degree of success with few side-effects and its good bioavailability [12,16,17].

We aimed to report our real-life experience of *C. acnes* shoulder implant joint infection (SIJI) management in a French university hospital in which clindamycin-based treatment is commonly used. 

## 2. Results

### 2.1. Characteristics of the Population

From 2010 to 2019, the clinical orthopedic records system Orthoplus^®^ recorded 1270 procedures for reverse total shoulder arthroplasty (RTSA), 426 for anatomic total shoulder arthroplasty (ATSA), and 221 for hemiarthroplasties, and from 2015 to 2019, 504 revision procedures for osteosyntheses. Among those, 148 patients had *C. acnes* positive samples. We excluded nine polymicrobial infections. Finally, 48 monomicrobial *C. acnes* SIJI were identified. The average follow-up was 45 months.

The characteristics of the population are presented in Table 1. Among the 48 SIJI, we found 47 (98%) late chronic SIJI, 0 delayed SIJI, and 1 (2%) early postoperative SIJI. A total of 27/48 (56%) patients had undergone a mean number of one other surgical procedure (other than the index arthroplasty) on the infected joint. These were mainly arthroplasties (n = 15) or surgery without any implant such as rotator cuff repair or dislocation surgeries (n = 12). No patient had previous subacromial injection.

### 2.2. Surgical Management of the Population

Concerning the 30 infected RTSA, 29 (97%) revisions were performed because of a high suspicion of infection, with 21 (72%) two-stage revisions, debridement and implant retention (DAIR) for four (14%) patients, three (10%) one-stage revisions, and one (3%) drainage of a peri-prosthetic abscess. One case (3%) was an open reduction of dislocation.

For the eight infected ATSA, six revisions were performed because of a high suspicion of infection and two because of mechanical issues. There were six one-stage revisions and two two-stage revisions. 

For the five infected hemiarthroplasties, two patients had a preoperative suspicion of infection and three patients had a diagnosis of infection during revision for RTSA implantation on osteoarthritis. There were three one-stage revisions, one two-stages revision, and one arthroscopic tendon repair.

For the five infected osteosyntheses, all revisions were performed for mechanical problems, with four one-stage revisions and one arthroscopy for nerve compression.

When a two-stage revision of the device was performed, the median delay between the two stages was 12 weeks [5,6,7,8,9,10,11,12,13,14,15,16,17,18,19,20,21,22,23,24,25,26,27,28,29,30,31,32].

### 2.3. Microbiology Results and Resistance Profile

Note that 3/48 (6%) patients had one positive sample for *C. acnes* out of several samples taken during placement of the index implant. All the isolates tested for amoxicillin + clavulanic acid, piperacillin with tazobactam, cyclines, or rifampicin were susceptible. We found one clindamycin-resistant strain (2%) from a patient who had not received this agent previously. Among the six cases with a previous PJI at the same site, four patients (67%) had been treated previously with clindamycin and 100% had a clindamycin susceptible *C. acnes* strain associated with the SPJI. The susceptibility profiles of the *C. acnes* strains to the different antibiotics are presented in Table 2.

Six patients out of 48 (13%) cases had a previous history of BJI on the same joint, three with *C. acnes*, two with CoNS of, which one was associated with *S. aureus* and *Streptococcus mitis*, and one patient lacking microbiological documentation. While the rate of favorable outcome in patients without history of BJI was 35/42 (83%), this rate fell to 3/6 (50%) in patients with a previous history of BJI. 

### 2.4. Antibiotic Therapy

Twenty-nine patients were treated with monotherapy, 25 (86%) with clindamycin, three (10%) with amoxicillin, and one (4%) with amoxicillin and clavulanic acid.

Sixteen patients were treated with dual therapy including fluoroquinolone (n = 13), clindamycin (n = 9), rifampicin (n = 7), and penicillin (n = 6). The most frequent combination was clindamycin with fluoroquinolone for 6/16 cases (38%). 

The mean (±SD) duration of antibiotic treatment was 6.57 ± 2 weeks. Three patients had surgery without associated antibiotic treatment.

### 2.5. Treatment Outcome According to Clindamycin Use

Among the 33 patients treated with clindamycin, 25 (76%) received monotherapy and eight (24%) received dual therapy, with a median treatment duration of 6 weeks (6–12).

The use of clindamycin in the context of mono or dual therapy had no significant impact on the outcome (Figure 1). The difference between treatment with or without clindamycin did not have an impact on prognosis (Figure 2). No side effects requiring clindamycin discontinuation were recorded.

Among patients treated with clindamycin, surgical treatment consisted of two-stage revision for 17/33 (52%) patients, one-stage for 11/33 (33%) patients, one DAIR for 3/33 (9%) patients, drainage of a peri-prosthetic abscess for 1/33 (3%) patients, and no revision for 1/33 (3%) patients.

Twenty-seven patients out of 33 (82%) were cured with clindamycin and the rate increased to 27/31 (87%) when failures due to lack of septic revision strategies were excluded (one abscess drainage without DAIR and one open reduction of dislocation) (Table 3). The cure rate of patients treated with antibiotics other than clindamycin was 9/12 (75%), increasing to 9/10 (90%) when the no surgical revision strategy was excluded (arthroscopy for nerve compression and DAIR on chronic infection) (*P* = 1.0).

Two of the six failures were associated with a *C. acnes* strain resistant to clindamycin isolated during a new revision surgery. One failure occurred with a *C. acnes* strain displaying the same antibiotype (i.e., clindamycin-susceptible). Three failures occurred with different bacteria. If we assume that the failure occurred with the same strain, the incidence rate of emerging clindamycin resistance after clindamycin treatment is 6%.

## 3. Discussion

Clindamycin was used in 65% of *C. acnes* SIJI treated at the University Hospital of Nice between 2011 and 2019. In the present study, we report a favorable outcome without relapse in 87% of patients treated with clindamycin when a revision strategy was applied, with an equivalent efficacy of clindamycin compared to other strategies mainly based on fluoroquinolone with or without rifampicin. No clindamycin discontinuation because of adverse effects was required.

Studies concerning antibiotic therapy for *C. acnes* PJI are rare, especially those focusing on shoulder protheses [8,18,19,20,21]. Jacobs et al. found a favorable outcome rate of 88% for 60 SPJI, among which 49 were treated with clindamycin with or without rifampicin [19]. Zeller et al. reported a 92% therapeutic success for 50 *C. acnes* PJI, six of which were SP; twenty-one were treated with an association of clindamycin and rifampicin [18]. Vilchez et al. reported a 72% therapeutic success rate with five patients treated with clindamycin alone and six with clindamycin and rifampicin [21]. Despite the fact that clindamycin is proposed in the guidelines as an alternative choice for *C. acnes*, in practice, it is often used as a first-line treatment with a satisfactory rate of success [18,19,21]. Clindamycin monotherapy has already shown its efficiency in other clinical situations, particularly in hidradenitis suppurativa [22,23]. In BJI, Jacobs et al. reported the use of clindamycin monotherapy in prosthetic joint infections in 16 patients and recorded two failures [19]. The use of clindamycin as a monotherapy precludes the interaction with rifampicin [24,25]. 

In the study of Zeller et al., patients were treated for a mean duration of 16 weeks with oral antibiotics (including clindamycin + rifampin) after a mean duration of 5 weeks with intravenous antibiotics (including clindamycin + rifampin) [18]. Concerning the study of Vilchez et al., the median duration of antibiotic therapy was 56 days, with the use of clindamycin in 11/44 cases [21], which is closer to ours (mean ± SD: 6.57 ± 2 weeks). In the literature, we found a median follow-up of 24 months in the study of Piggott et al. and an average follow-up of respectively 540 days for Pradier et al., and at least 1 year for Jacobs et al. [8,19,26]. The average follow-up of 45 months reported in our study reinforces the outcome classification provided.

The *C. acnes* isolates found in device-related infections were shown to form a greater amount of biofilm compared to isolates from healthy skin [27,28]. Rifampicin is known to be active against bacteria producing biofilms in PJI, notably against *S. aureus* [14]. The results of the addition of rifampicin together with other antibiotics in an in vitro experiment model demonstrated that the anti-biofilm action of rifampicin provided a better rate of resolution of *C. acnes* infection than monotherapy without rifampicin [29,30]. However, the exact impact of Rifampicin for *C. acnes* PJI still has to be determined. Our results are in accordance with other studies that did not find an adjunctive effect of rifampicin on patients’ outcomes [8,19], but more recently, a retrospective European study with a significantly higher number of patients included (187 patients) reported a 0.5 adjusted hazard ratio for treatment failure that favors dual therapy with rifampicin [31]. Despite not being statistically significant, this result deserves further evaluations.

The data in the literature concerning the emergence of clindamycin-resistant strains of *C. acnes* following treatment are heterogeneous, varying from 6 to 30% [5,32,33]. It is difficult to analyze the treatment failure of SIJI due to clindamycin-resistant strains isolated from patients for whom the strain was initially susceptible to this molecule; due to the lack of a genotyping method, we are unable to differentiate acquired resistance induced by clindamycin treatment from an infection with a different strain.

The way in which the surgery is performed depends primarily on chronological criteria. In the case of a delayed or late infection, the surgical strategy should consist in exchange of the implant [34,35]. The recommendations do not clearly outline the choice of one- or two-stage revision, which follows the usual practices of the center [9,18,36]. One-stage replacement has the potential advantage of leading to fewer post-operative complications and improved functional outcome [6,37]. One-stage revision is limited by the type of prosthesis used and depends on the integrity of the rotator cuff [9,38]. Management of the group “no revision strategies” or the delayed DAIR in our reference orthopedic center is explained by the difficulty of managing these *C. acnes* implant joint infections, especially when infection is obvious (i.e., fistula), but function is preserved. 

In our study, two of three SIJI were successfully treated without medication, one with surgical replacement of the implant and one with arthroscopic tendon repair on hemiarthroplasties. These successful results obtained without antibiotic treatment raise the hypothesis of *C. acnes* strains of variable pathogenesis. The phylogeny of *C. acnes* has been little studied. Phylotype IA is considered to dominate in moderate to severe acne, while the phylotypes Ib, II. and III are most prevalent in infections of soft tissues and medical implants [39]. El Sayed et al. provided a new perspective of the definition of infection or culture contamination depending on whether the *C. acnes* isolates belonged to a distinct or similar ST [40]. The classical criteria used to diagnose PJI may be undermined by the underestimated complexity of the microbiological diagnosis, but also because of a specific histological presentation. In fact, classical polymorphonuclear neutrophil infiltration can be missing. The reference center of bone and joint infections in Lyon, France reported substantial plasmocytic infiltration associated with *C. acnes* PJI in 71.4% of patients [41]. 

The main limitations of our study are represented by the variability of the antibiotic strategies and surgical strategies applied to our patients and the study’s small simple size. Including patients with a history of SIJI could be seen as a limitation, since they are more at risk for failure at baseline, thus leading to an underestimation of clindamycin’s efficacy. 

## 4. Materials and Methods

### 4.1. Study Design and Population 

This was a retrospective observational single center study performed at the University Hospital of Nice in France. Inclusion criteria were: One shoulder surgical procedure (including arthroscopy) performed between 2011 and 2019 in patients with a history of shoulder implants RTSA, ATSA, hemiarthroplasties, and osteosyntheses;At least one intraoperative sample positive for *C. acnes* in culture. We excluded all other types of infection and primary surgery.

These patients were included from the clinical orthopedic records system Orthoplus^®^. Then, the clinical data were crossed with the microbiological database of our University Hospital (Glims^®^) to select patients with positive intraoperative samples for *C. acnes*. Only monomicrobial *C. acnes* SIJI were analyzed.

### 4.2. Follow-up and Treatment Outcome

We established the *C. acnes* SIJI definition according to French [34] and international recommendations [9,35] based on: at least two peri-implant positive *C. acnes* cultures (obtained from two distinct samples) or at least one peri-implant positive *C. acnes* culture associated with a peri-prosthetic fistula. Infections were classified according to IDSA guidelines: early postoperative infection (<3 months after index surgery), delayed infection (between 3 and 12 months after index surgery), and late infection (>12 months after index surgery). The primary endpoint was: favorable outcome of *C. acnes* SIJI treated with clindamycin, defined as at least 1-year follow-up free of infection symptoms without suppressive antibiotic treatment. Patient outcome was defined as favorable if the initial symptoms cleared without proceeding with other surgical revision, or in the case of a sterile bacteriological sample obtained during another procedure, after antibiotic treatment. Other patients were considered as failures. The therapeutic strategy was based on multidisciplinary decision making including experts in shoulder surgery (PB) and ID specialist dedicated to PJI management (JC) [42].

### 4.3. Data Collection

The epidemiological, clinical, biological, and therapeutic data were recovered in a standardized manner from patients’ electronic and paper medical records. The demographic data included age, sex, comorbidities, and treatments such as anticoagulation and corticotherapy. The type of infected implants was specified: ATSA, RTSA, hemiarthroplasties, and osteosyntheses. The reason for the implant revision was included. The delay between implantation of the device and SIJI was recorded along with the number and type of surgery performed before infection, and if subacromial injection had been performed. The biological data included markers of inflammation with C-reactive protein and leucocyte and polymorphonuclear neutrophil counts. Clinical data included symptoms such as pain and signs of local inflammation. The surgical procedures, type, and duration of antibiotic treatment were recorded.

### 4.4. Antibiotic Susceptibility Tests

The antibiotic susceptibility tests were performed using the disk diffusion method according to the European Committee on Antimicrobial Susceptibility Testing (EUCSAT) for aerobic bacteria and according to the Comité de l’antibiogramme de la société française de microbiologie (CA-SFM) 2013 for obligate anaerobes [43]. The clindamycin minimal inhibitory concentration was determined by E-test^®^ (bioMérieux, Marcy l’Etoile, France) with a MacFarland 1 standard inoculum, incubated under anaerobic conditions for 48 to 72 h and interpreted according to EUCAST for the strains classified as resistant by the disk diffusion method.

### 4.5. Clindamycin Treatment

Clindamycin was given orally 600 mg q.i.d. for weights over 70 kg and t.i.d. for weights equal to or less than 70 kg.

### 4.6. Statistical Analysis

If not specified, continuous variables are expressed as medians and interquartile ranges, and categorical variables are expressed as counts and valid percentages. Statistical analyses were performed using GraphPad Prism 7 software. Fisher or χ^2^ tests were used for comparison between groups for non-parametric qualitative variables when appropriate. *p*-value significancy was set at 0.05.

## 5. Conclusions

In conclusion, the therapeutic strategy based on a one-stage or two-stage revision together with 6 weeks of clindamycin appears to be effective and safe for SIJI involving *C. acnes*. However, clindamycin should be used with caution if this agent has been previously administered.

## Figures and Tables

**Figure 1 antibiotics-11-00608-f001:**
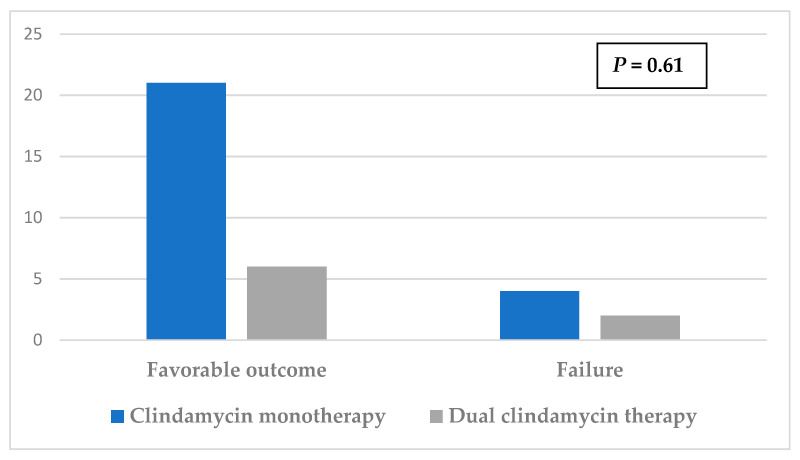
Effect on prognosis of mono versus dual clindamycin therapy.

**Figure 2 antibiotics-11-00608-f002:**
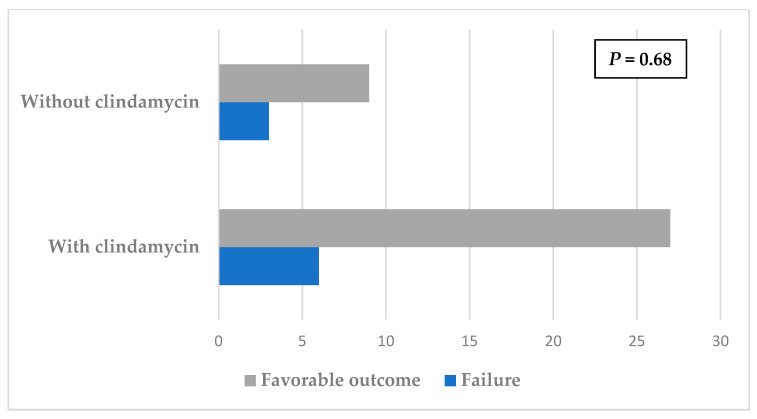
Impact of clindamycin therapy on prognosis.

**Table 1 antibiotics-11-00608-t001:** Demographic, clinical, and biological characteristics of patients with shoulder device-related *C. acnes* infections.

Variable	n = 48
Age, years	67 (12)
Male	36 (75%)
Diabetic	14 (29%)
Obesity IMC > 30 kg/m^2^	1 (2%)
Coronaropathy	6 (13%)
Cirrhosis	1 (2%)
Rhumatoid polyarthrites	1 (2%)
Anticoagulant	3 (6%)
Corticotherapy	1 (2%)
RTSA	30 (63%)
ATSA	8 (17%)
Osteosynthesis	5 (10%)
Hemiarthroplasty	5 (10%)
Time since index arthroplasty, months	24 (49.5)
Number of previous interventions, mean ^a^	1
History of BJI at the same site	6 (13%)
Local inflammation	7 (15%)
Shoulder pain	42 (88%)
C-reactive protein [mg/L]	22.59
Leucocytes [G/L]	7.99
Polymorphonuclear neutrophiles [G/L]	5.71

**^a^** Other than the index arthroplasty.

**Table 2 antibiotics-11-00608-t002:** Susceptibility of the 48 strains of *C. acnes* to the different antibiotics of therapeutic interest.

Antimicrobial Agent	Rate of Susceptible Strains (%) *
Beta-lactam + beta lactam inhibitor	48/48 (100)
Moxifloxacin	40/40 (100)
Rifampicin	8/8 (100)
clindamycin	47/48 (98)
Tetracycline	48/48 (100)
Linezolid	44/44 (100)

* Susceptibility test results were not available for all the antibiotics.

**Table 3 antibiotics-11-00608-t003:** The favorable outcome rate of the different therapeutic strategies associated to treatment with clindamycin.

Medical and Surgical Treatment	Rate of Favorable Outcome (%)
Monotherapy + 2-stage surgery	10/12 (83)
Dual therapy + 2-stage surgery	4/5 (80)
Monotherapy + 1-stage surgery	11/11 (100)
Dual therapy + DAIR	2/3 (67)
Monotherapy + Abscess drainage	0/1 (0)
Monotherapy + Absence of revision	0/1 (0)

## Data Availability

Data are available upon reasonable request.

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
