# Peer review of "Clindamycin Efficacy for Cutibacterium acnes Shoulder Device-Related Infections"

_antibiotics, 2022, doi:10.3390/antibiotics11050608_

Round 1
Reviewer 1 Report
The paper ,,Clindamycin efficacy for Cutibacterium acnes shoulder device-related infections,, is of clinical significance. Please, consider my suggestions.
1. Please, write the name of the species of bacteria e.g. Cutibacterium acnes and C. acnes in italics,
2. clindamycin should be written in lowercase,
3. line 50 The susceptibility profile of C. acnes also allows 50
the use of fluoroquinolones, - The susceptibility profile of C. acnes in our clinic also allows the use of fluoroquinolones. Point that it applies to your clinic.
4. Explain the acronym when you use it in the text for the first time (RTSA, ATSA, IPJI).
5. Table 1. Explain: Age (mean, years) 65 (24,85). 65 is a mean age? What does 24.85 mean? Please specify the age range.
Use range instead of extrema.
Locale inflammation, correct- Local inflammation.
Write the names of the metric units in square brackets e.g. [mg/L]
Figure 1. Correct: Succes and Echec (is it in Freanch?)
Figure 2. Correct: Succes
Please, what does Succes mean? Explain this criterion.
Table 2. What does ,,Total Nb (%),, mean?
Line 215: monocentric, Please, correct, single center study.
Author Response
The paper, Clindamycin efficacy for Cutibacterium acnes shoulder device-related infections, is of clinical significance. Please, consider my suggestions.
1. Please, write the name of the species of bacteria e.g. Cutibacterium acnes and C. acnes in italics,
Corrected in the whole manuscript
2. clindamycin should be written in lowercase,
Corrected in the whole manuscript
3. line 50 The susceptibility profile of C. acnes also allows 50
the use of fluoroquinolones, - The susceptibility profile of C. acnes in our clinic also allows the use of fluoroquinolones. Point that it applies to your clinic.
Added
4. Explain the acronym when you use it in the text for the first time (RTSA, ATSA, IPJI).
Added the first time it’s used line 62 and 68
5. Table 1. Explain: Age (mean, years) 65 (24,85). 65 is a mean age? What does 24.85 mean?
Please specify the age range.
Use range instead of extrema.
Continues variables are presented as median + IQR
Locale inflammation, correct- Local inflammation. Done
Write the names of the metric units in square brackets e.g. [mg/L] Done
Figure 1. Correct: Succes and Echec (is it in Freanch?)
Favorable outcome and failure are now used
Figure 2. Correct: Succes
Done
Please, what does Succes mean? Explain this criterion.
Favorable outcome is defined now line 246
Table 2. What does ,,Total Nb (%),, mean?
Title of the column is now: Rate of susceptible strains (%)
Line 215: monocentric, Please, correct, single center study.
Corrected line 226

Reviewer 2 Report
The study aimed to evaluate the efficacy of clindamycin, an antibiotic in guidelines for c. acnes PJI, in the treatment of BJI to C. acnes.
The objective is very interesting, but I have a strongly problem of methodology.
To evaluate antibiotic efficacy, the authors aimed to compare treatment outcome with or without clindamycin but population and C. acnes PJI are different (there are polymicrobial co infections (9/57), and mono infections, patients who had a previous history of PJI and patients without previous problems...). Patients and their BJI are so differents to compare efficacy of clindamycin.
A randomized prospective study is able to answer to this objective. I advice authors to revise their objective.
Author Response
Reviewer 2
The study aimed to evaluate the efficacy of clindamycin, an antibiotic in guidelines for c. acnes PJI, in the treatment of BJI to C. acnes.
The objective is very interesting, but I have a strongly problem of methodology.
To evaluate antibiotic efficacy, the authors aimed to compare treatment outcome with or without clindamycin but population and C. acnes PJI are different (there are polymicrobial co infections (9/57), and mono infections, patients who had a previous history of PJI and patients without previous problems...). Patients and their BJI are so differents to compare efficacy of clindamycin.
A randomized prospective study is able to answer to this objective. I advice authors to revise their objective.
We acknowledge a part of heterogeneity in the type of shoulder implant joint infections and their associated management presented in the work.
In order to improve results presentation, we have excluded polymicrobial infections (see Materials and methods line 231). Only monomicrobial C. acnes SIJI are now presented in the manuscript.
Management of patients with a history of PJI failure is common and we don’t think we should exclude them for such work. Nevertheless, we took into consideration this comment while describing line 115 the number of favorable outcomes in the 6 patients with a history of PJI of the same joint. As excepted, it is lower than the rest of the population. One may point out that it could lead to an underestimation of clindamycin efficacy in patients with a first PJI. However, nobody can argue that it is a misleading way to present our result to the readers.
Our objective is to share a real-life experience of clindamycin use for SIJI. The last sentence of the introduction which summarize our objective has been changed line 62.
We acknowledge that RCT (i.e., a different objective) are required: we have submitted a proposal for a French national funding in clinical research for a multicentric protocol studying C. acnes PJI 3 months ago.

Reviewer 3 Report
I read, Clindamycin efficacy for Cutibacterium acnes shoulder device related infections, with interest. In this manuscript, the authors aimed to evaluate the efficacy of Clindamycin in the treatment of C. acnes shoulder implant joint infections (SIJI).
I have some questions and suggestion.
1) Can you explain why this study is new or telling new things?
2) Discussion is rather weak. The data from other studies is relatively small. Please add more efficacy of Clindamycin-based treatment for C. acnes from the other studies as well for comparison with this study.
3) Please add more limitation in your study. For example, small sample size.
4) Please provide more data on the importance of physicians and pharmacy around the world to recognize Clindamycin-based treatment for C. acnes.
Minor:
Line 15: Please remove “Background:”
Author Response
Reviewer 3
I read, Clindamycin efficacy for Cutibacterium acnes shoulder device related infections, with interest. In this manuscript, the authors aimed to evaluate the efficacy of Clindamycin in the treatment of C. acnes shoulder implant joint infections (SIJI).
I have some questions and suggestion.
1) Can you explain why this study is new or telling new things?
We do not pretend to provide a brand-new way to manage SIJI. Clindamycin is an old drug. However, data regarding its use for SIJI can be considered as extremely limited: among the references discussed in our paper n°8 reports 24 infections, n°18 reports 6 infections, n°19 reports 9 infections and n°20 reports 12 infections.
2) Discussion is rather weak. The data from other studies is relatively small. Please add more efficacy of Clindamycin-based treatment for C. acnes from the other studies as well for comparison with this study.
We added the data of other studies especially for duration of treatment and time of follow-up in order to confirm the consistency of our results Line 170-178. The last Spanish study published in Antibiotics is also discussed now.
3) Please add more limitation in your study. For example, small sample size.
We added the sample size and the fact that we included 6 patients with a history of PJI (see comment of reviewer 2)
4) Please provide more data on the importance of physicians and pharmacy around the world to recognize Clindamycin-based treatment for C. acnes.
We are sorry but we are not sure to understand this comment.
Regarding the “data”: we are not able to provide more efficacy or safety results in our cohort.
Regarding the “rational” of clindamycin use in SIJI: we have already highlighted the limits and drawbacks of the 2 potential comparators (type A penicillin + fluoroquinolones) and the potential companion of these drugs (rifampin) line 47 to 56 in the introduction section. Qualities of clindamycin (i.e. bone diffusion and bioavailability) are described line 57 to 61. This rational by itself is considered from our point of view as relevant.
Minor:
Line 15: Please remove “Background:”
Done
